# Influence of anthropogenic emissions on the composition of highly oxygenated organic molecules in Helsinki: a street canyon and urban background station comparison

Magdalena Okuljar[1], Olga Garmash[2*], Miska Olin[2], Joni Kalliokoski[2], Hilkka Timonen[3], Jarkko V. Niemi[4], Pauli Paasonen[1], Jenni Kontkanen[1**], Yanjun Zhang[1***], Heidi Hellén[3], Heino Kuuluvainen[2], Mina Aurela[3], Hanna E. Manninen[4], Mikko Sipilä[1], Topi Rönkkö[2], Tuukka Petäjä[1], Markku Kulmala[1], Miikka Dal Maso[2] and Mikael Ehn[1]

[1]Institute of Atmospheric and Earth System Science, Faculty of Science / Physics, Faculty of Science, University of Helsinki, FI-00014, Helsinki, Finland
[2]Aerosol Physics Laboratory, Physics Unit, Tampere University, PO Box 692, FI-33014, Tampere, Finland
[3]Atmospheric Composition Research, Finnish Meteorological Institute, Helsinki, Finland
[4]Helsinki Region Environmental Services Authority, HSY, PO Box 100, FI-00066, Helsinki, Finland
[*]Now at the Department of Atmospheric Sciences, University of Washington, Seattle, WA, United States
[**]Now at CSC – IT Center for Science Ltd., Espoo, Finland
[***]Now at Univ. Lyon, Université Claude Bernard Lyon 1, CNRS, IRCELYON, 69626 Villeurbanne, France

*Correspondence to*: Magadalena Okuljar (magdalena.okuljar@helsinki.fi)

**Abstract.** Condensable vapors, including highly oxygenated organic molecules (HOM), govern secondary organic aerosol formation and thereby impact the amount, composition, and properties (e.g. toxicity) of aerosol particles. These vapors are mainly formed in the atmosphere through the oxidation of volatile organic compounds (VOCs). Urban environments contain a variety of VOCs from both anthropogenic and biogenic sources, as well as other species, for instance nitrogen oxides ($NO_x$), that can greatly influence the formation pathways of condensable vapors like HOM. During the last decade, our understanding of HOM composition and formation has increased dramatically, with most experiments performed in forests or in heavily polluted urban areas. However, studies on the main sources for condensable vapors and secondary organic aerosols (SOA) in biogenically influenced urban areas, such as suburbs or small cities, has been limited. Here, we studied the HOM composition, measured with two nitrate-based chemical ionization mass spectrometers and analyzed using positive matrix factorization (PMF), during late spring at two locations in Helsinki, Finland. Comparing the measured concentrations at a street canyon site and a nearby urban background station, we found a strong influence of $NO_x$ on the HOM formation at both stations, in agreement with previous studies conducted in urban areas. Even though both stations are dominated by anthropogenic VOCs, most of the identified condensable vapors originated from biogenic precursors. This implies that in Helsinki anthropogenic activities mainly influence HOM formation by the effect of $NO_x$ on the biogenic VOC oxidation. At the urban background station, we found condensable vapors formed from two biogenic VOC groups (monoterpenes and sesquiterpenes), while at the street canyon, the only identified biogenic HOM precursor was monoterpenes. At the street canyon, we also observed oxidation products of aliphatic VOCs, which were not observed at the urban background station. The only factors that clearly correlate (temporally and composition-wise) between the two stations contained monoterpene-derived dimers. This suggests that HOM composition and formation mechanisms are strongly dependent on localized emissions and the oxidative environment in these biogenically influenced urban areas, and they can change considerably also within distances of one kilometer within the urban environment.

## 1. Introduction

Urban environments can contain various anthropogenic and biogenic sources of volatile organic compounds (VOCs). Biogenic emissions come mostly from urban vegetation, for example, trees and bushes in parks, gardens, and may contain biogenic volatile organic compounds (BVOCs) such as isoprene, monoterpenes (MT), or sesquiterpenes. The sources of anthropogenic emissions consist of traffic, industrial processes and production of customer goods, and volatile chemical products (VCP) (Li et al., 2022; Koppmann, 2007; Watson et al., 2001). Gas-phase compounds emitted from anthropogenic sources contain trace gases, including nitrogen oxides ($NO_x$), as well as anthropogenic volatile organic compounds (AVOCs), for example aromatic compounds or aliphatic hydrocarbons (Timonen et al., 2017; McDonald et al., 2018). In densely populated areas, VCPs can dominate AVOCs concentrations and compounds typically known as BVOC (e.g., monoterpenes) are also emitted from anthropogenic sources, such as personal care products and cleaning agents (Gkatzelis et al., 2021; Li et al., 2022).

Under atmospheric conditions, VOCs can undergo oxidation to form condensable vapors (Pandis et al., 1992; Ehn et al., 2014). The most common ambient oxidants are ozone ($O_3$), hydroxyl radical (OH), and nitrate radical ($NO_3$) (Wayne, 2000). $O_3$ is a trace gas produced in the troposphere mostly by photolysis of $NO_2$ (Liu et al., 1980), and present in the ambient air during the entire day. $O_3$ can oxidize only VOC containing at least one double or triple bound, or, with a slower reaction rate, carbonyls (Bianchi et al., 2019). OH is a short-lived, highly-reactive compound produced mostly by the photolysis of $O_3$ (Crutzen et al., 1999), thus OH is present in the atmosphere mainly during the daytime. $NO_3$ is a product of the reaction between $O_3$ and $NO_2$, which gets rapidly destroyed by photolysis and reactions with NO during the daytime (Wayne et al., 1991). Both radicals can react with most closed-shell VOCs (Seinfeld and Pandis, 2016), but in the atmosphere, $NO_3$ reacts mostly with alkenes while OH reacts with almost all compounds, including aromatic hydrocarbons (Seinfeld and Pandis, 2016). Oxidation of VOCs almost always leads to peroxy radical ($RO_2$) intermediates, typically with long enough lifetimes to participate in bimolecular reactions, primarily with NO, $HO_2$, or other $RO_2$. The $RO_2$ may also undergo various unimolecular isomerizations, and both these and the bimolecular reactions can lead to either propagation or termination of the organic radical (Bianchi et al., 2019). The structure of the final product depends on multiple factors, including the structure of the initial VOC and the "oxidative conditions", meaning available oxidants and the bimolecular reaction partners. The latter can be referred to as "terminators" when they terminate the oxidation process, and in some cases the product composition can tell a lot about the oxidative conditions. Additionally, $NO_2$ can terminate oxidation chain in reaction leading in most cases, which decompose back to substrates (Atkinson and Arey, 2003). For example, $RO_2$ termination by NO and oxidation by $NO_3$ can produce organic nitrogen compounds (ONCs), organonitrates (Atkinson and Arey, 2003; Bianchi et al., 2019), while $RO_2$ termination by $NO_2$ can form relatively unstable peroxy nitrates. $RO_2$ cross reactions are the only reactions that can form accretion products, ROOR, referred to here as "dimers" (Valiev et al., 2019).

$RO_2$ intermediates can also undergo autoxidation, where the $RO_2$ isomerizes through a hydrogen shift (H-shift) creating an alkyl radical to which molecular oxygen can attached (Bianchi et al., 2019; Ehn et al., 2014; Crounse et al., 2013). In the end, a new, more oxidized $RO_2$ is formed, which can either undergo additional H-shifts or bimolecular reactions, with both potentially terminating or propagating the oxidation (Bianchi et al., 2019) chain. In cases where the radical can undergo multiple autoxidation H-shifts, the end product can reach high enough oxidation levels to be classified as HOM (Bianchi et al., 2019). The structure of a VOC strongly influences its propensity to undergo autoxidation and, consequently, the molar yield of HOM. This results in the very variable HOM yields, which can reach high values for different anthropogenic and biogenic compounds (Molteni et al., 2018; Bianchi et al., 2019; Garmash et al., 2020).

Differences in the structural composition affect both the physical and chemical properties of HOM, with more oxidized products typically being less volatile (Kroll and Seinfeld, 2008). However, the exact functionalities are important, and e.g. oxygen atoms in nitrate groups lower the volatility much less than if the oxygen was found in some other functional group (Kroll and Seinfeld, 2008). In general, the high oxygen content of HOM makes them an important contributor to secondary organic aerosol (SOA) formation, influencing e.g. air quality.

During the last decade, HOM formation from biogenic emissions have been extensively studied in forests (Ehn et al., 2014; Yan et al., 2016; Bianchi et al., 2017; Massoli et al., 2018), and in agricultural environments (Kürten et al., 2016). Recently, research showed that also the oxidation of AVOCs can noticeably contribute to the HOM population (Molteni et al., 2018; Garmash et al., 2020; Wang et al., 2021) and SOA formation (Timonen et al., 2017). Additionally, $NO_x$ can alter the HOM formation mechanism and influence SOA formation (Fry et al., 2014; Ng et al., 2017; Pullinen et al., 2020; Mutzel et al., 2021). Due to these findings, the research on condensable vapors and their origin focused stronger on urban environments, especially very polluted ones, heavily influenced by anthropogenic emissions (Brean et al., 2019; Liu et al., 2021; Guo et al., 2022b; Nie et al., 2022; Yan et al., 2022). In very polluted environments, formation of condensable vapors is greatly impacted by $NO_x$ (Brean et al., 2019; Liu et al., 2021; Guo et al., 2022b; Nie et al., 2022; Yan et al., 2022) and HOM composition is often dominated by AVOC precursors (Nie et al., 2022).

While the composition and formation of condensable vapors have been studied in the above-mentioned forests and highly polluted locations, environments with considerable influence from both anthropogenic and biogenic emission sources have received much less attention. Such areas include urban environments with lots of green areas, for example suburbs, or cities surrounded by large forests. A better understanding of such locations may also help to assess the impact on air quality from adding vegetation such as green roofs to already built-up areas. Helsinki is an example of a city with forests in close proximity, and Saarikoski et al. (2023) estimated that there, even at a street canyon site strongly affected by traffic emissions, BVOCs are the main contributor to oxidation products. While Saarikoski et al. (2023) measured only the composition of VOCs, and not their oxidation products, this finding makes us expect that the relative role of BVOCs is even higher for HOM, as BVOCs typically have higher propensity for autoxidation than AVOC (Bianchi et al., 2019). Another important aspect to consider is the spatial representativeness of typical urban measurements. As cities are very inhomogeneous in terms of local emissions and the oxidative environment, and HOM are short-lived compounds, HOM studies in urban environments that were performed at one specific location may not be comparable to other nearby locations with different urban sub-environments.

Here we investigate the composition of condensable vapors at two nearby stations in Helsinki, which are differently influenced by anthropogenic emissions. The first station is located in a busy street canyon while the second is in an urban background area, at less densely built part of Helsinki, 150 meters from the nearest busy road. We studied the composition of condensable vapors, mostly HOM, at these sites using two nitrate-based chemical ionization mass spectrometers. To identify different HOM types from the mass spectra and connect them to different formation pathways, we applied Positive Matrix Factorization (PMF) to separate co-varying species. We compared the drivers of HOM formation between the two urban sub-environments and explored the roles of biogenic and anthropogenic emissions on HOM composition, in order to understand how these can affect the air quality in urban environments with a strong biogenic influence.

## 2. Methods

We measured the composition of condensable vapors at two stations in Helsinki situated in contrasting environments: the Helsinki Region Environmental Services Authority (HSY) air quality station (60°11´47.0´´ N, 24°57´07.7´´ E) and the Station for Measuring Ecosystem-Atmosphere Relations (SMEAR III, 60°12´10.4´´ N, 24°57´40.2´´ E) (Fig. 1). The HSY supersite is located at a street canyon, less than a meter from Mäkelänkatu street (around 28 000 vehicles/weekday) (Kuuluvainen et al., 2018). SMEAR III is 900 m north-east of the HSY station and with 150 m distance from the closest busy road (Hämeentie street). SMEAR III, is classified as an urban background station (Järvi et al., 2009). The neighborhood of these stations was previously described in Okuljar et al. (2021). Here we refer to them as "street canyon" (later also "SC") and "urban background station" (later also abbreviated "UB"), respectively.

The measurement campaign was conducted during 11 May 2018 – 03 June 2018 at the urban background station and 27 April 2018 – 24 May 2018 at the street canyon. The measurement period was during change of season and by 14 May 2018 the deciduous trees in the surrounding area had fully developed their leaves. To study the influence of traffic emissions, we analyzed separately the data measured during workdays as well as only during weekends and public holidays (1 May 2018 and 10 May 2018). We refer to them as 'workdays' and 'weekends', respectively. As the nighttime concentrations are often influenced by the emission from the previous day, we separate these categories in 24 h periods starting at 4 a.m. The corresponding analysis of size distribution of 1-1000 nm particles measured during the same time at both stations in Helsinki is presented by Okuljar et. al. (2021).

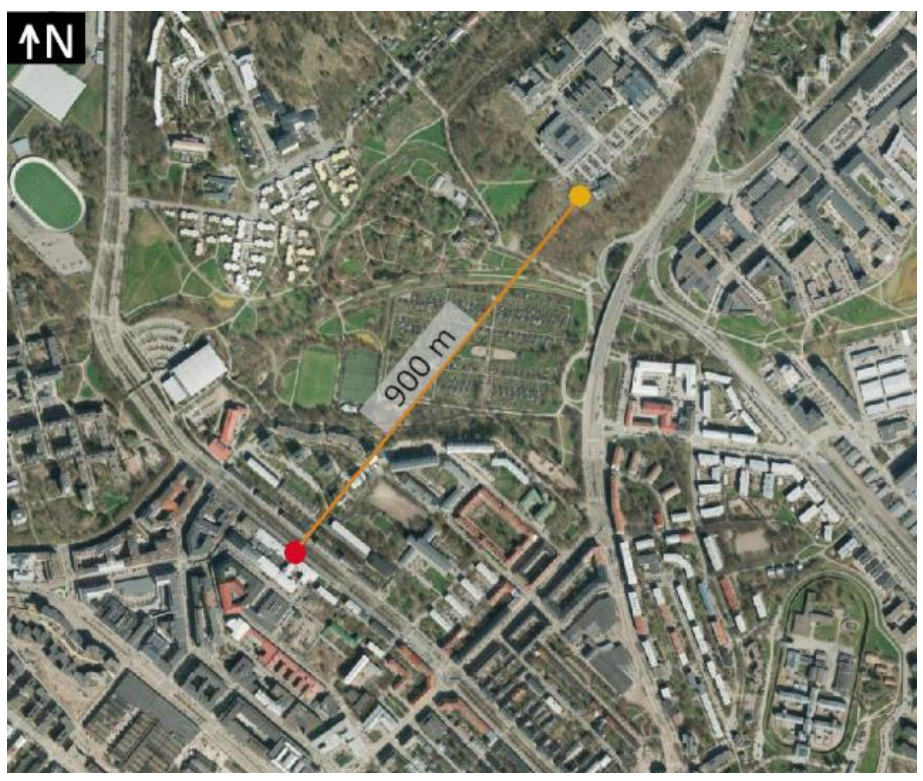

Figure 1. Orthophotograph of stations in street canyon (red) and in an urban background environment (yellow) made on May 7$^{th}$, 2018. The photograph was provided by The City of Helsinki map service (CC BY 4.0).

### 2.1. Condensable vapor measurements

The composition of condensable vapors was measured simultaneously at both stations by two nitrate-ion based chemical ionization atmospheric pressure interface (CI-APi-TOF) mass spectrometers (MS) (Jokinen et al., 2012). Nitrate ions ($NO_3^-$), produced by interactions between soft x-ray and sheath air containing nitric acid ($HNO_3$), binds to the analyzed compound through hydrogen bonds or charges the analyte via proton transfer reactions. $NO_3^-$ is primarily selective towards organic molecules containing at least two suitably positioned hydroxyl (-OH) or hydroperoxyl groups (-OOH) (Hyttinen et al., 2015), or compounds with higher gas-phase acidity than $HNO_3$. After the sample gets ionized, the ions are focused in the APi module and ultimately separated in the time-of-flight (TOF) analyzer based on their mass-to-charge ratio (m/Q, reported in units of Th). The CI-APi-TOF and its working principle was described in detail by Jokinen et al.

(2012). The resolving power of the MS at both stations was approximately 3000-4000 Th/Th for signals with m/Q higher than 200 Th. The mass spectra were analyzed using the software package tofTools (Junninen et al., 2010).

In measured mass spectra, we observed multiple peaks at every m/Q. To perform high-resolution (HR) analysis requires us to fit closely set signals and could increase uncertainties of results. Therefore, most of our analysis is based on unit mass resolution (UMR) data and HR analysis follow only when we can narrow it usage. Additionally, we noted that condensable vapor measurement at street canyon had lower transmission for higher m/Q than at urban background station.

Here, we discuss quantitative changes in condensable vapors based on their measured signal in counts per second (cps) normalized by the cps of the reagent ions, using the unit ncps (normalized cps). The ambient concentrations can be estimated by using previously determined instrument-specific calibration coefficients for sulfuric acid (Okuljar et al., 2021) equal to $4 \cdot 10^9$ $cm^{-3}$ for the street canyon station and $7 \cdot 10^9$ $cm^{-3}$ for the urban background station. However, usage of these calibration coefficient determined for sulfuric acid to calculate HOM concentration comes with very large uncertainties, and we therefore concentrate on comparison of ion signal strength.

### 2.2. VOC measurements

VOC concentrations were measured at the street canyon with an offline method in which ambient samples were first collected on a Tenax TA-Carbopack B sorbent tube and later analyzed by thermal desorption gas chromatography coupled with mass spectrometry (TD-GC-MS). We measured VOC concentrations during the period 15 – 25 May 2018 with 4 h time resolution. 13 analytes were classified as AVOCs: benzene, toluene, ethylbenzene, p/m-xylene, styrene, o-xylene, propylbenzene, 3-ethyltoluene, 4-ethyltoluene, 1,3,5-trimethylbenzene, 2-ethyltoluene, 1,2,4-trimethylbenzene, and 1,2,3-trimethylbenzene, and 15 as BVOCs: monoterpenoids (α-pinene, camphene, β-pinene, Δ3-carene, p-cymene, 1,8-cineol, limonene, terpinolene), terpene alcohol (linalool), an oxidation product of β-pinene (nopinone), bornyl acetate, and sesquiterpenes (longicyclene, iso-longifolene, β-caryophyllene and α- farnesene). More detailed decription of the method can be found e.g. in Helin et al. (2020).

### 2.3. Other instrumentation

$CO_2$, NO, $NO_2$, $SO_2$ as well as meteorological variables were measured at both stations. Table S1 contains information about measurements of additional variables used in this paper.

### 2.4. Positive Matrix Factorization

Collected datasets from the measurement of condensable vapors at both stations consist of an enormous amount of information and it is challenging to filter data that contain relevant information for analysis of HOM formation. As both stations are located in a city, the composition of condensable vapors is dependent on different types of VOC sources as well as chemical and metrological conditions. To extract relevant information, separate different pathways of HOM formation, and find processes affecting condensable vapor composition at both stations, we applied Positive Matrix Factorization (PMF) (Paatero, 1997; Paatero and Tapper, 1994; Paatero and Hopke, 2003). PMF is a multivariate factor analysis model which has been widely used on aerosol mass spectrometry data (Ng et al., 2011; Zhang et al., 2011; Chen et al., 2022) and more recently on ambient gas-phase chemical ionization mass spectrometry data (Yan et al., 2016; Massoli et al., 2018; Zhang et al., 2019; Liu et al., 2021; Nie et al., 2022).

We performed PMF analysis on three different m/Q ranges from UMR data at both stations: 200-350 Th, 350-500 Th, and 500-650 Th. In this paper, we will refer to these ranges as ranges 1, 2, and 3, respectively. The loss rate of HOM due to condensation is roughly a function of their mass (Peräkylä et al., 2020), thus, analyzing mass spectra in ranges allows us to group HOM with similar loss rates and focus specifically on separating the HOM sources (Zhang et al., 2020). Additionally, when a m/Q range has lower signal than other ranges, it will only have a minor weight on the PMF solution and relevant information may be lost (Zhang et al., 2020). Using m/Q ranges for PMF analysis is important especially at the street canyon as it may partly counteract the loss of information due to lower transmission for higher m/Q. The focus of our analysis is on compounds in a range of 200 to 650 Th as in this reach we can find majority of the condensable vapors containing $C_{5\text{-}20}$. Smaller m/Q are unlikely to condense, while larger m/Q had very low, or even negligible, signals. We prepared data and error matrices with 30 min time resolution, separately for each range at each station according to the methods described by Yan et al. (2016). To conduct PMF analysis we used the Igor-based interface Source Finder (SoFi, version 6.D) (Canonaco et al., 2013) and ME-2 solver (Paatero, 1999). Detailed information about data preparation and validation of PMF solutions can be found in S1.

To describe the chemical composition of ions in obtained factors, we determined the times for each factor when that factor had the highest relative contribution to the total signal and then fit peaks to the HR data to identify the key compounds. Choosing times when the analyte is dominant across all factors in the same m/Q range and at the same site is necessary to ensure that identified compound is correctly assigned to the factor. In this paper, we performed a more detailed interpretation only of chosen factors from each station, which we refer to further on as "selected factors". A factor was chosen for further interpretation only when we could reasonably identify ions in it, and relate it to a real atmospheric source, i.e. not impurities. We refer to other factors as "not selected factors". Examples of each type will be given later to better clarify this selection process.

### 2.5. Limitation of data for interpretation

The are several limitations for interpreting the data. At the street canyon, a low signal is observed for higher m/Q. That leads to a low signal-to-noise ratio (S/N) for HOM measured in range 2 and 3, and in some cases makes it impossible to identify the compounds. As a result of fast decrease of measured signal with an increase of m/Q, at the street canyon, over 90 % of the signal of 200-650 Th is located in range 1. This could be caused by the low transmission for higher m/Q of the CI-APi-TOF measurement at that station. Transmission is a results of voltage settings in CI-APi-TOF, which are optimized for each instrument separately. Zha et al. (2018) showed that the ratio of the signal for the same sampled air measured by two CI-APi-TOF can change drastically with an increase of m/Q due to the difference in the transmission between instruments. Thus, the highest uncertainty caused by inconsistent transmission between two instruments is observed in range 3. Nevertheless, this uncertainty does not influence identification of peaks that have sufficient S/N.

Due to the chemical complexity of the samples, we cannot achieve high accuracy of mass calibration on some of the measured days. This is the reason why we have performed PMF analysis on UMR data. Limitations of peak identification due to the MS resolution and the presence of multiple overlapping peaks also hinder the identification of some ions, and hence we are confident to report only the dominant ions in each factor. We are not able to report key compounds for factors that have minor contribution to their m/Q range or have too many similar peaks with other factors, as we cannot unambiguously assign identified compounds to these specific factors.

Lastly, we need to keep in mind that chemical ionization with $NO_3^-$ is very selective, mostly towards highly functionalized compounds. Overall, this ionization method is optimal for detection of HOM, however, it limits observations of other oxidation products.

# 3. Results and discussion

We start this section by providing a short overview of the meteorological conditions during the campaign. In section 3.2, we present our main findings, starting from the PMF results and the subsequent interpretations of important formation pathways of condensable vapors at the two measurement sites. In the last part of this section, we discuss the potential implications of our findings on the air quality in Helsinki.

Concerning notations, we focus our study on HOM, but we also detect abundant organic compounds which contain less than six oxygen atoms, which do not classify as HOM. Thus, we often use a broader term 'condensable vapors' when discussing observed products more broadly. In addition, we observe monomeric (mostly $C_9$-$C_{10}$) and dimeric (mostly $C_{19}$-$C_{20}$) oxidation products of MT, which we refer to as 'MT-derived monomers' and 'MT-derived dimers', respectively. For simplicity, we call factors containing monomeric oxidation products of MT 'MT monomers' while 'MT dimers' factors contain of dimeric oxidation products of MT.

## 3.1. Overview of meteorological and trace gas conditions in Helsinki

Atmospheric conditions, for example local emissions and oxidative environment, influence HOM formation pathways. To understand HOM formation mechanisms and their differences between studied sites, we first investigated meteorological and chemical conditions at both stations. Figure 2 presents diurnal variations of measured variables that can influence HOM formation pathways: global radiation, ambient temperature (T), and concentrations of $O_3$, NO, and $NO_2$. As mentioned earlier, the measurement periods overlapped but were not identical between the two stations. Therefore, differences in campaign averages between sites are partly driven by differences in location and partly by differences in time.

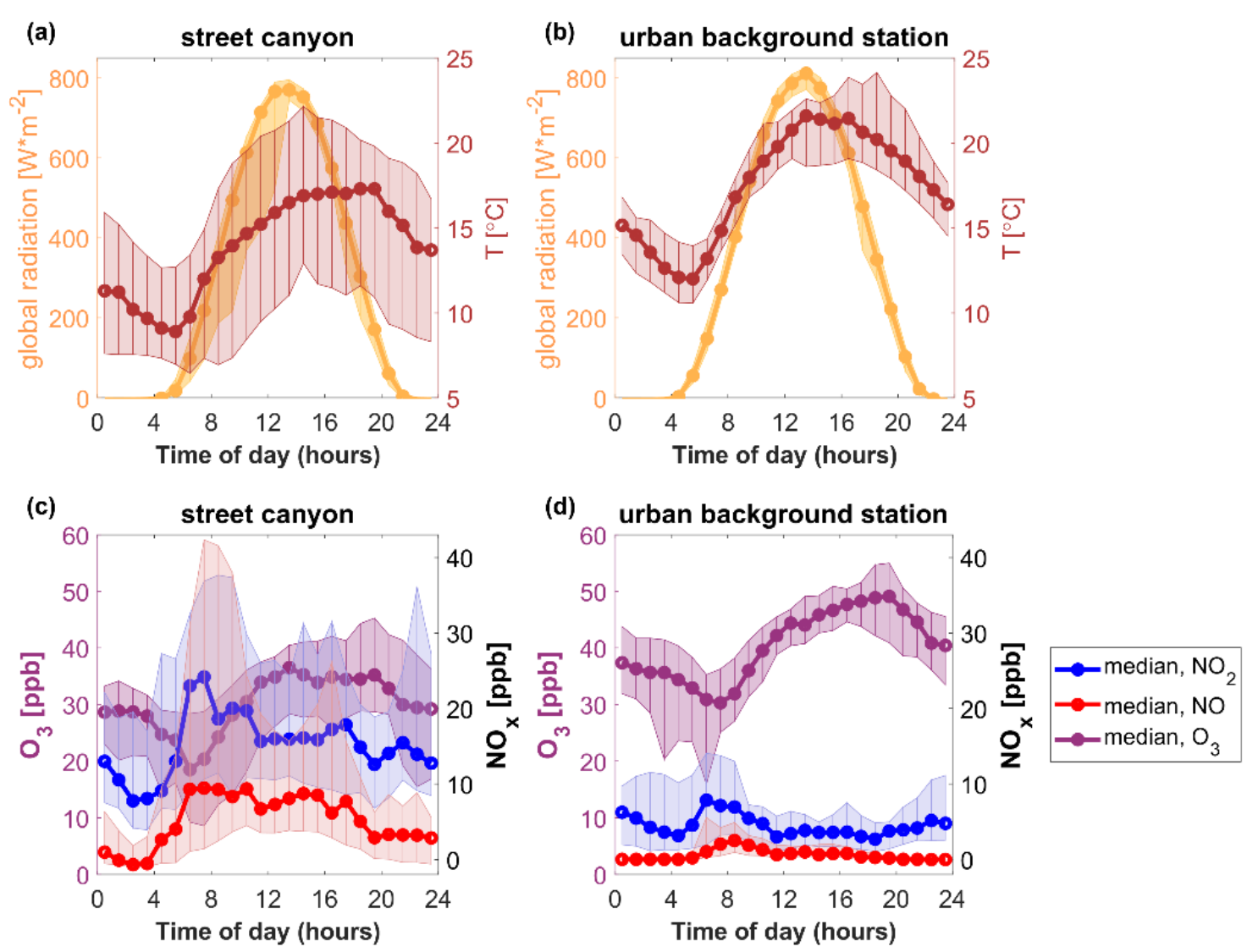

Figure 2. Diurnal variations of (a,b) global radiation and ambient temperature, and (c,d) NO, $NO_2$, and $O_3$ concentrations at the street canyon (left) and urban background station (right). Presented data contain both workdays and weekends. The median diurnal variations are shown as solid lines with markers; 25$^{th}$ to 75$^{th}$ percentile ranges are presented as shaded areas. Time is local.

Diurnal variation of global radiation is similar between the two stations (Figure 2a,b), though with slightly more cloudy periods at the street canyon. Global radiation initiates photolysis reactions and, as a result, enhances the formation of OH and $O_3$ as well as the decomposition of $NO_3$. Median temperatures varied between 12.0°C and 21.6°C at the urban background site and between 8.9°C and 17.3°C in the street canyon. Higher temperature at the urban background station can be explained by the difference in measurement periods as the measurements started two weeks later than in street canyon. During the period when measurements overlapped, the median temperature is very similar between stations reaching almost 22°C during daytime and dropping to 12-13.4°C during nighttime (figure not shown). The increase in temperature typically accelerates molecular reaction rates as well as enhances BVOCs emissions and evaporation rates. It can also affect HOM yields (Quelever et al., 2019).

At the urban background station, NO has a maximum between 8:00 and 9:00 (2.5 ppb) and it is negligible during nighttime. In contrast, at the street canyon, the median NO concentration was below the detection limit between 1:00 and 3:00, after which it rapidly increased, levelling off at 7:00 and staying elevated (ca. 9 ppb) throughout the day until 17:00. That means NO can affect oxidation reactions more at the street canyon site, even during much of the night, when it stays at 4 ppb until early morning. In the context of VOC oxidation, the presence of NO likely causes the termination of the oxidation. In the absence of NO, termination reactions with $RO_2$ become more favorable. $NO_2$ and NO (Figure 2c,d) concentrations are up to 5 and 23 times higher at the street canyon than at the urban background station, respectively. At urban background site, $O_3$ reaches minimum median concentration at 7:00 (30.3 ppb) and maximum at 19:00 (49.1 ppb). At the street canyon, the corresponding values are 18.5ppb at 6:00 and 36.5 ppb at 13:00. During overlapping times between the sites, median $O_3$ concentration stays 5-25 ppb lower at street canyon than at urban background station (figure not shown). It could be partly associated with higher NO concentration in street canyon as NO reacts with $O_3$. $O_3$ remains relevant for VOCs oxidation throughout the day. $O_3$ and $NO_2$ concentrations affect production of $NO_3$ and thus its concentration.

### 3.2. Characterization of PMF factors

In this subsection, we examine the HOM composition and formation at both stations by investigating PMF factors in all three m/Q ranges (200-350 Th, 350-500 Th, 500-650 Th); we focus our analysis on selected factors, their time-series, and diurnal variations (Fig. 3-4) as well as mass spectra (Fig. S2-3). We refer to PMF factors as SCX-Y or UBX-Y where SC is the street canyon, UB is the urban background station, X is the analyzed m/Q range (either 1, 2, or 3), and Y is the identifying number of the factor in that range. The factors also appear together with a descriptive name. As an example, "*UB3-2: MT dimers*" refers to the second PMF factor identified in mass range 3 at the urban background site and was found to mainly contain ions related to monoterpene-derived dimers. To understand the chemical composition of factors, we identify their key compounds with HR data (Table S2) as described in Sect. 2.5. All key compounds are detected as clusters with $NO_3^-$ or $HNO_3NO_3^-$ and this is how we report them in Table S2 and on the mass spectra (Fig. S2-3 and S6-7); however, for clarity of the interpretation in this subsection, we write their chemical structures without the nitrate adducts.

The PMF analysis involves several lengthy steps, including determining an optimal number of factors in the solution, as well as interpreting sources for each factor based on the supporting data available. In this study, we had six data sets to analyze (two sites and three mass ranges). As a result, we summarize the key characteristics of each factor and give an interpretation in the main text and present more detailed description of the PMF analysis in the supplementary information (SI) (Sect. S2 motivates the choice of factor numbers, Sect. S3 describes the main features of each factor, which lead to the interpretations given in the main text). In the following sections, we first briefly describe the overall characteristics of factors observed at the street canyon (Sect. 3.2.1) and at the urban background station (Sect. 3.2.2) after which we compare HOM formation and composition between these sites (Sect. 3.2.3).

Only "selected" factors are described here, while characteristics of "not selected" factors are presented and described in the SI (Sect. S3, Fig. S6-7, Table S3). Several reasons motivated us not to select factors for detailed discussion in the main text. For example, a factor was not selected if it was a contamination or an artefact (e.g., containing mainly water clusters isotopes) or if we were not confident in the meaningful separation of this factor by PMF method (this was the case for the entire range 1 at the street canyon, as described below). Overall, we selected 5 out of 13 factors from the street canyon and 10 out of 14 factors from the urban background station. These selected factors explain 34%, 100%, and 100% of the observed signal in ranges 1-3 at the urban background station and 0%, 64%, and 61% of the observed signal in ranges 1-3 at the street canyon, respectively (Table 1, Fig. S3-4).

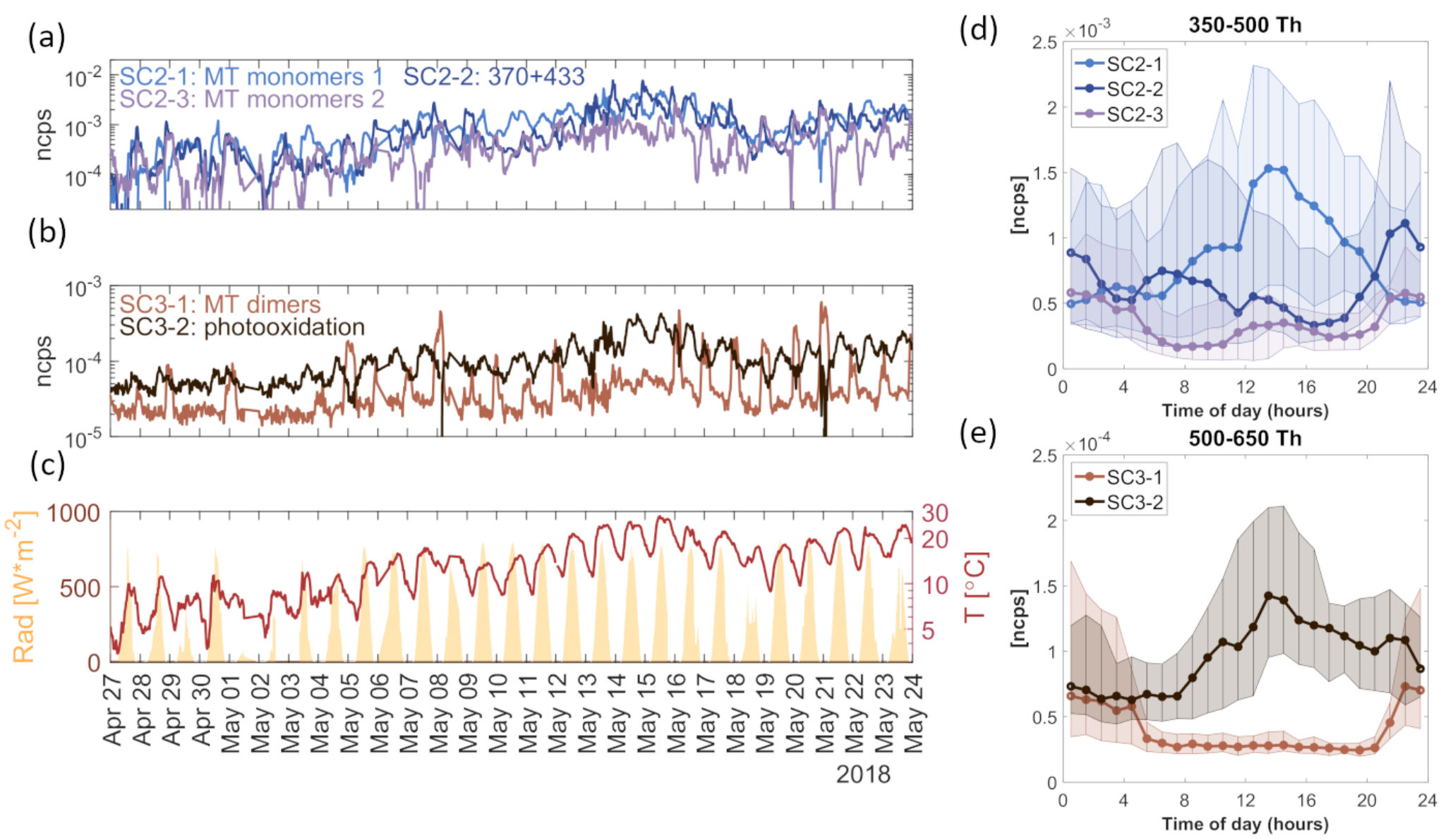

Figure 3. Time series of selected PMF factors (a-b), global radiation and ambient temperature (c), and diurnal variation of PMF factors fractions (d-e) at street canyon (SC). The median diurnal variation is shown as a solid line with markers; 25th to 75th percentile ranges are presented as shaded areas. Y-axis in ncps indicates the measured signal in normalized counts per second.

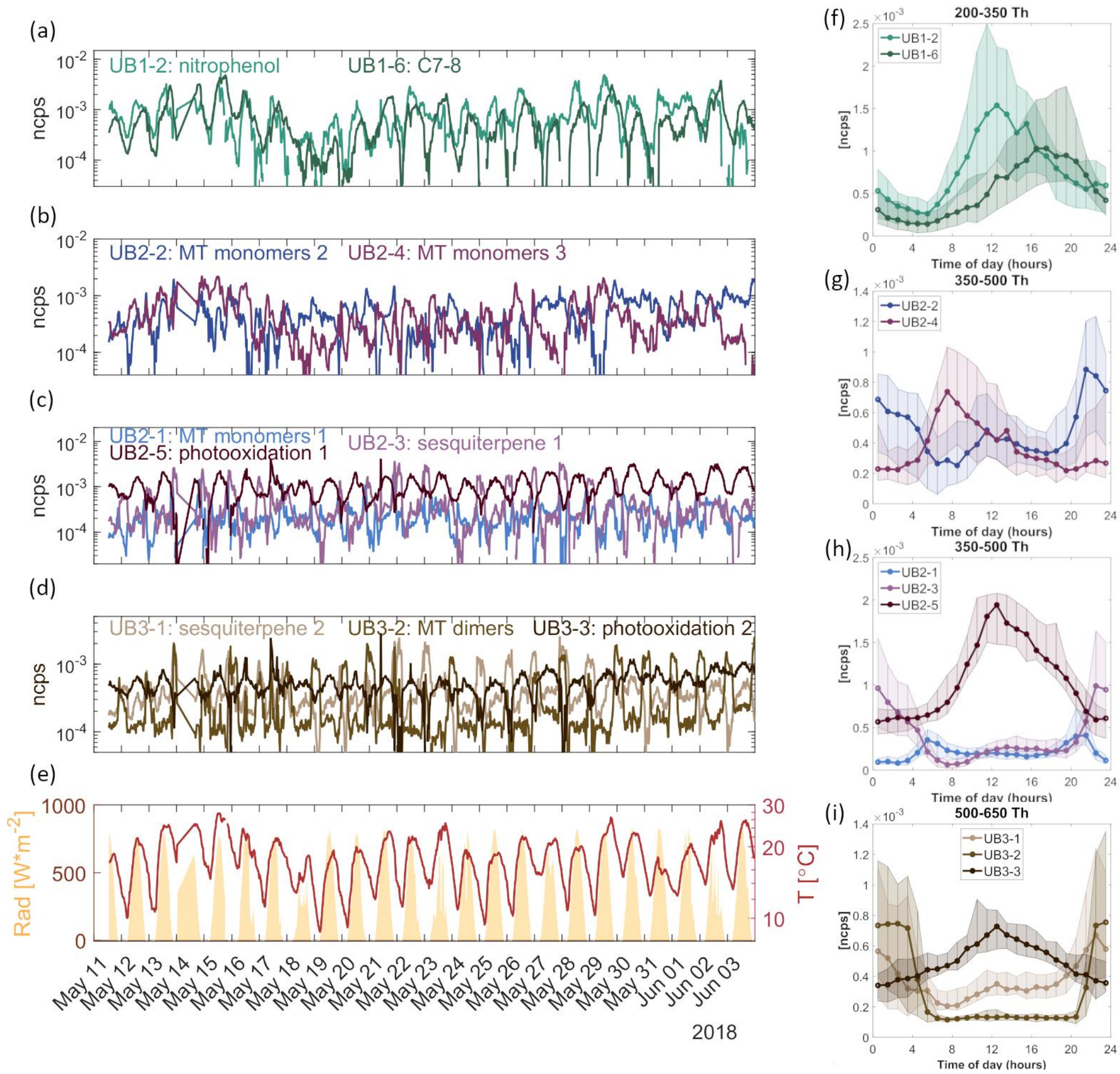

Figure 4. Time series of selected PMF factors (a-d), global radiation and ambient temperature (e), and diurnal variation of PMF factors (f-i) at urban background station (UB). The median diurnal variation is shown as a solid line with markers; the 25th and 75th percentile ranges are presented as shaded areas. Y-axis in ncps indicates the measured signal in normalized counts per second.

### 3.2.1. Street canyon

Here, we very briefly describe factors observed at the street canyon site in each m/Q range. The main examination of the factors is given in Sect. 3.2.3 where we also discuss them in relation to factors observed at the urban background station. As indicated already above, PMF solutions for range 1 at the street canyon were inconclusive, and therefore all factors from this range are classified as 'not selected'. The main reason is that all factors had very similar temporal trends, mainly correlating with temperature. This may be a result of most observed molecules being semi-volatile, and increased temperatures lead to increased evaporation of these molecules. In any case, as PMF relies on temporal variability to separate factors, too much co-variance makes PMF less reliable. Nevertheless, we believe there was some useful

information also in this range and will briefly discuss *SC1-1: nitrophenol 1*, *SC1-2: MT monomers 3*, and *SC1-5: nitrophenol & aliphatic* in this section.

**Range 1, 200-350 Th (Factors selected: 0/5)**

In range 1, all factors are affected by changes in ambient temperature (Fig. S6 and S10). Factors in range 1 have a daytime peak and nearly all of them could have been oxidized by OH or $O_3$ (Fig. S6, Table S3). Most of these factors are likely formed from AVOCs and contain nitrophenol ($C_6H_5O_3N$) as well as other N-containing aromatics, such as nitrocresol. Nitrophenol can be directly emitted from combustion or formed from benzene and phenol oxidation. The presence of nitrophenol in many factors can be explained by an abundance of benzene at the street canyon as it is the third most abundant VOC measured at the street canyon site (Fig. S8).

**Range 2, 350-500 Th (Factors selected: 3/5)**

In range 2, all selected factors respond to the changes in the ambient temperature (Fig. 3 and S10), especially factor SC2-1, which contains monomeric oxidation products of MT (MT-derived monomers) with nitrate functionalities. Factors SC2-2 and SC2-3 are highest during the night, but they also have local maxima during the day, which suggests that competing processes influence the formation of these factors and thus their diurnal pattern. SC2-3 may be inhibited by NO, as it decreases when NO reaches its daily maximum (Figure 3d). SC2-3 consists of MT-derived monomers, while SC2-2 is dominated by one single compound: $C_{10}H_{16}O_9N_2$.

**Range 3, 500-650 Th (Factors selected: 2/3)**

Range 3 contains one daytime and one nighttime factor. SC3-1 is a MT-derived dimer factor produced via oxidation by $NO_3$ and present during the night, when NO concentrations are low enough to allow $RO_2$ termination via $RO_2$ cross reactions. SC3-2 is a daytime factor containing HOM oxidized by OH. SC3-2 also has some signal from instrumental impurities containing fluorine (F-impurities), and undefined noise peaks.

#### 3.2.2. Urban background station

Similar to the previous subsubsection, we describe briefly factors observed in all ranges at the urban background station. The discussion about these factors follows in Sect. 3.2.3, in which we compare factors found at both sites in Helsinki.

**Range 1, 200-350 Th (Factors selected: 2/6)**

Selected factors in range 1 contain daytime factors, from which UB1-6 is a factor correlating the best with the ambient temperature (Fig. 4 and S1). Time-series of UB1-6 correlates with $O_3$ (Fig 4a, Fig. S11) and it contains key compounds with $C_{7\text{-}8}$ atoms. These formulas have been detected earlier as products of MT oxidation in chamber studies (Yan et al., 2020) and in ambient measurement (Liu et al., 2021), however, they have also been identified as oxidation products of aromatic VOCs (Guo et al., 2022b). Since CI-APi-TOF does not provide information about molecular structure, we cannot unambiguously determine the origin of this factor. In contrast to UB1-6, UB1-2 factor contains nitrophenol and likely originates from AVOCs. The diurnal variation of UB1-2 resembles the one expected for OH (Saarikoski et al., 2023). Both UB1-2 and UB1-6 contain ONCs and their oxidation was likely terminated by NO or $NO_2$.

**Range 2, 350-500 Th (Factors selected: 5/5)**

Range 2 contains various daytime and nighttime factors (Fig. 4). Factor UB2-1 reaches the highest concentrations at 5 am. and 10 pm., which corresponds to the time of sunrise and sunset during our measurement period. As this factor consists of MT-derived HOM with two N-atoms, we can speculate that they are formed from $NO_3$ oxidation of MT and terminated by NO. It is typically assumed that $NO_3$ and NO would not co-exist. However, simultaneous presence of $NO_3$ and NO when photolysis is just high enough to form NO but not to fully deplete $NO_3$ is a plausible explanation for the diurnal pattern of UB2-1.

UB2-2 and UB2-3 are both nighttime factors oxidized mainly by $NO_3$ and inhibited by NO during daytime. UB2-2 contains MT-derived monomers and correlates with the MT-derived dimer factor (UB3-2). Following the diurnal cycle in Fig. 4g, it can be observed that when the concentration of UB2-2 decreases, the concentration of daytime MT-derived monomer factor, UB2-4, increases. Even though UB2-2 and UB2-4 both contain key compounds with $C_{9-10}$, the molecular formulas are slightly different. Specifically, in UB2-2 key compounds contain one or three N-atoms while in UB2-4 they have zero or two N-atoms. UB2-2 and UB2-4 could thus be formed from competing HOM formation pathways from the same VOCs.

In contrast to other factors, UB2-3 consists of HOM with composition of $C_{15}H_{23}O_{8,10-16}N$, based on which we conclude that this factor is formed from sesquiterpenes ($C_{15}H_{24}$, Richters et al., 2016). UB2-3 correlates very well with a corresponding sesquiterpene factor from range 3, UB3-1 (R=0.93) (Fig. S9). The last factor UB2-5 is a daytime factor which during noon corresponds to more than 50% of the measured signal (Fig. S5). It is most likely that UB2-5 is formed in OH oxidation.

**Range 3, 500-650 Th (Factors selected: 3/3)**

Range 3 at UB site contains two nighttime factors: sesquitepene-derived UB3-1 factor, and MT-derived dimer UB3-2 factor. Both factors consist of ONCs, products of $NO_3$ oxidation of BVOCs, and are inhibited by NO, being absent during the day as a result. UB3-3 is the only daytime factor (Fig. 4i) in range 3 and it consists of OH-oxidized HOM, F-impurities, and noise.

### 3.2.3. Factor interpretation and comparison between urban background and street canyon sites

Tables 1 and S3 present the most plausible interpretation of selected and not selected factors, respectively. For each factor, we propose VOC precursors, oxidants, and terminators, which were most likely to influence the formation of species in this factor. We also specify an hour of the day when factor's signal reached its maximum as well as the contribution of this factor to the total signal both within its own m/Q sub-range and within the full analyzed range (200-650 Th). See Table 1 caption for a more detailed description of how to read the table. The findings and implications are discussed below.

Table 1. Suggested characterization of selected factors at both stations. Detailed factor interpretation is described in Sect. S3. The importance of the various species described in this table was assessed based on either factor time series (TS), factor mass spectra (MS), or both (B), as indicated by the superscript in the "Factor" column. The "Precursor" column describes which type of molecules we expect to act as precursors to the observed signals, separating (when possible) between AVOC and BVOC. The "Oxidant" and "Terminator" columns depict our estimates for the most likely species involved in the oxidation process ("M", as in "maybe", is used if we were unable to exclude or confirm the participation of the species). If the "yes" or "no" is marked in bold font, it means that we found a particularly clear influence of that species for that factor. The "Diurnal peak time" shows the hour when the factor had its highest concentration, and "Fraction" depicts the percentage of signal (of the given sub-range or the total analyzed m/Q range) that the factor contributed to.

| Range [Th] | Factor | Precursor | Oxidant | | | Terminator | | | Diurnal peak time | Fraction [%] within | |
|---|---|---|---|---|---|---|---|---|---|---|---|
| | | | OH | $NO_3$ | $O_3$ | NO | $RO_2$ | $HO_2$ | | 200-650 | Sub-range |
| **Street canyon** | | | | | | | | | | | |
| | SC2-1$^{TS}$ | BVOCs | M | no | M | yes | no | M | 13 | 2.2 | 27.2 |
| **350-500** | SC2-2$^{TS}$ | VOCs | no | **yes** | M | M | no | M | 22 | 2.0 | 24.8 |
| | SC2-3$^{TS}$ | BVOCs | no | **yes** | no | no | M | M | 0 | 0.9 | 11.5 |
| **500-650** | SC3-1$^{B}$ | BVOCs | no | yes | no | **no** | yes | no | 22 | 0.1 | 19.3 |
| | SC3-2$^{MS}$ | VOCs, noise, F-impurities | **yes** | no | M | yes | no | M | 13 | 0.2 | 41.6 |
| **Urban background station** | | | | | | | | | | | |
| **200-350** | UB1-2$^{MS}$ | AVOCs | **yes** | no | no | yes | no | M | 12 | 8.9 | 16.1 |
| | UB1-6$^{TS}$ | VOCs | M | no | M | yes | no | M | 17 | 7.4 | 17.8 |
| | UB2-1$^{TS}$ | BVOCs | no | **yes** | no | yes | no | M | 21 | 2.6 | 8.6 |
| | UB2-2$^{TS}$ | BVOCs | no | yes | M | **no** | M | M | 21 | 5.4 | 18.0 |
| **350-500** | UB2-3$^{MS}$ | BVOCs | no | yes | M | **no** | no | M | 22 | 5.1 | 17.0 |
| | UB2-4$^{TS}$ | BVOCs | M | no | M | **yes** | no | M | 7 | 4.9 | 16.5 |
| | UB2-5$^{MS}$ | VOCs, noise | **yes** | no | M | yes | no | M | 12 | 12.0 | 39.9 |
| | UB3-1$^{B}$ | BVOCs | no | yes | M | **no** | no | M | 22 | 4.9 | 34.3 |
| **500-650** | UB3-2$^{B}$ | BVOCs | no | yes | M | **no** | yes | no | 23 | 3.6 | 25.4 |
| | UB3-3$^{MS}$ | VOCs, noise | **yes** | no | no | yes | no | M | 12 | 5.6 | 40.3 |

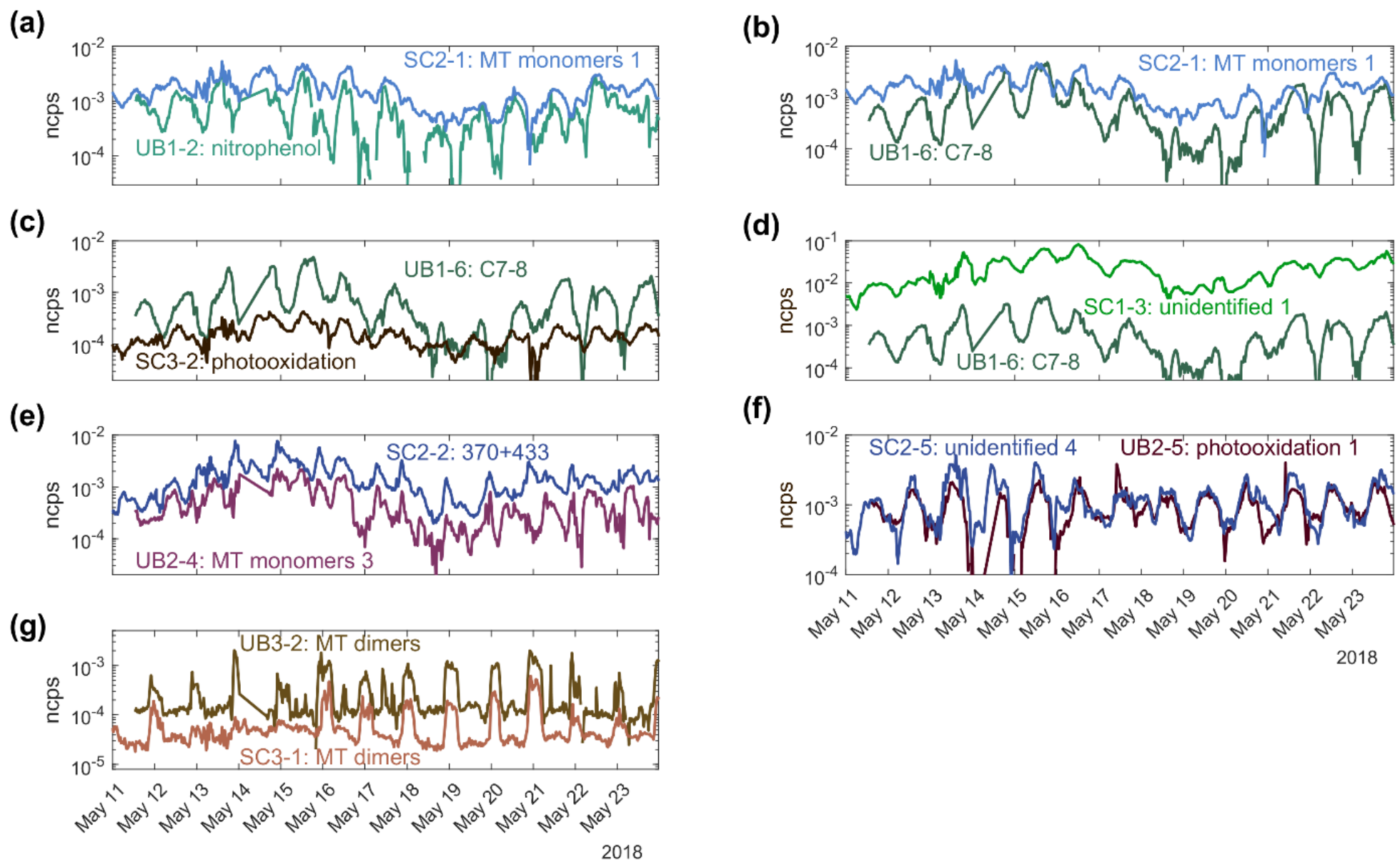

Figure 5. Time series of PMF factors with Pearson correlation coefficient higher than 0.7 (Figure S9) between the street canyon (SC) and the urban background station (UB) for common measurement time.

The total concentration of AVOCs is much higher than BVOCs at both stations (shown for street canyon in Fig. S8), however, most of the selected factors are more likely to have biogenic origin, primarily based on the identified peaks in the mass spectra (Table 1, Sect. S3). Our result is in agreement with the earlier study by Saarikoski et al. (2023) which concluded that, despite dominant AVOCs concentrations at street canyon site, BVOCs are estimated to be the main source of oxidized products, due to their higher reactivities. MT are the only type of biogenic precursors identified at the street canyon while at the urban background station we find oxidation products of both MT and sesquiterpenes. This is likely due to the difference in proximity of trees and vegetation from the stations as none of measured sesquiterpenes excided 0.2% of measured total BVOC concentrations at the street canyon. All key compounds detected in MT-derived factors were previously reported in studies investigating the influence of $NO_x$ on HOM formation from MT precursors (Pullinen et al., 2020; Yan et al., 2020; Shen et al., 2021; Dam et al., 2022; Guo et al., 2022a). Even though MT can be emitted from anthropogenic sources, for instance in form of VCPs (Gkatzelis et al., 2021; Li et al., 2022), the population density of Helsinki is low enough that that signature is likely lost among the abundant biogenic MT signals. This is in agreement with a-pinene being the most abundant BVOC and a common VCP, limonene (Coggon et al., 2021), being very low (<5% of BVOC concentration).

We observed only few PMF factors that we expect to originate from AVOC oxidation, and for both stations AVOC factors are detected only for the smallest m/Q sub-range, 200-350 Th (Table 1, Table S3). None of the factors had diurnal variations resembling traffic emissions at these sites (Olin et al., 2020; Okuljar et al., 2021). Even though concentrations of some factors differed between weekends and workdays, the diurnal behavior of factors was very similar (Fig. S12-13). This suggests that the emissions from traffic at these sites did not oxidize to form HOM at adequate yields or time scales to considerably contribute to our measured signals, in line with conclusions presented by Brean et al. (2019) and

Saarikoski et al. (2023). As precursor VOC concentrations are also affected by the mixing layer height (MLH), this effect may also impact HOM formation. However, VOCs are only one of the components that affect HOM formation, the effect of the MLH on HOM observed in this study is expected to be small.

Between the two stations, there are only a few factors with similar key compounds (Table S2). *SC3-1: MT dimers* has partly corresponding key compounds with *UB3-2: MT dimers*. They both contain $C_{20}H_{32}O_xN_2$ compounds, where x is 11-15 for SC and 13-16 for UB. *UB3-2: MT dimers* contains also other types of dimers which are not usually present in *SC3-1: MT dimers*. These slight differences between stations may be caused by the small difference in concentration of oxidant, or the concentration or type of MT. The non-negligible concentration of NO during nighttime at SC may also impact the dimer formation there. Nevertheless, these factors are very similar and form through similar pathways (Table 1) – oxidation of MT mostly by $NO_3$, and termination through $RO_2$ cross reactions, leading also to correlating time series (R=0.77., Fig. 5 and S9). The $RO_2$ + $RO_2$ reactions forming the MT-derived dimers will inevitably also form monomers as the dimer yield is never 100 %. However, monomers can also form through all other $RO_2$ termination channels, making them much more heterogeneous than the dimers. The time evolution of some MT-derived monomer factor time series (*SC2-3: MT monomers 2* and *UB2-2: MT monomers 2)* correlate with the corresponding dimer factors (R= 0.71 and R=0.67 respectively) as well as with each other (R=0.56). While both factors are dominated by $C_{10}$ compounds, their detailed mass spectra have significant differences (Fig. S2 and S3): *UB2-2: MT monomers 2* contains mainly ONCs with one N-atom while *SC2-3: MT monomers 2* has more compounds with two N-atoms (Table S2). This may indicate that there is enough NO available to terminate some fraction of the $RO_2$, yet without totally shutting down the $RO_2$ + $RO_2$ channel.

Another pair of factors showing similarities between the stations is *SC2-2: 370+433* and *UB2-4: MT monomers 3* (Fig. 5). Both factors are driven mostly by one compound ($C_{10}H_{16}O_9N_2$), which has been detected as two clusters $C_{10}H_{16}O_9N_2{\cdot}NO_3^-$ (370 Th) and $C_{10}H_{16}O_9N_2{\cdot}HNO_3{\cdot}NO_3^-$ (433 Th) in our instrument (determined by correlation analysis). The high time series correlation (R=0.75) suggests that molecules in these factors are formed via very similar pathways between the sites. Potentially, the formation pathways are identical, but importance of some competing pathways differ between the sites. Overall, the lack of stronger resemblance between these nearby sites suggests that even if HOM have the same VOC precursors, the environmental conditions regulate the relative importance between different oxidation pathways.

While differences in emissions and oxidation reactions will lead to diverse mass spectra, also the time series are expected to vary between the sites as the wind direction changes. For example, the street canyon site will likely be impacted by the street in different ways if the wind direction is from the street or towards the street. A clearly longer campaign than ours would be needed to identify the detailed impacts from different wind directions. However, analysis of the average diurnal variation can help us understand the roles of different oxidation conditions if the impact of varying wind directions diminishes in a longer average. Most factors at both stations can be characterized by one of a few types of diurnal patterns. Factors with a daytime diurnal variation reaching maximum concentration during noon or afternoon resemble diurnal variation of OH or $O_3$, respectively. However, temperature also peaks in the afternoon, and can lead to both higher BVOC emissions as well as evaporation of semi-volatile species from aerosols or surfaces, convoluting the effect of the oxidants on the observed HOM. Factors with noon or afternoon maxima are mostly found in range 1 at both sites, and to some extent in range 2. As these ranges mostly contain species thought to be semi-volatile (Peräkylä et al., 2020), it is possible that much of the observed variation is indeed due to the higher temperature causing increased partitioning of these

compounds into the gas phase. Nevertheless, OH and $O_3$ are likely involved as well, and given that the vast majority of signals are ONCs, $RO_2$ termination by NO is to be expected for most species. The opposite can be said for nighttime factors, which are likely inhibited in the daytime by NO, as their formation involves $RO_2$ termination via other pathways. This becomes especially visible for HOM terminated via $RO_2$ + $RO_2$ reactions (Ehn et al., 2014; Yan et al., 2016), which are mainly present in range 3. In this range, the volatilities are overwhelmingly low or extremely low, meaning that ambient temperature changes will not impact their ability to condense irreversibly to aerosols, thus also making their temporal behavior easier to interpret.

While daytime and nighttime peaks can be explained quite straightforwardly through variations in temperature or available oxidants or terminators that all follow distinct diurnal trends, we also observed additional types of diurnal trends, present mostly in range 2. Factor *UB2-1: Monoterpenes 1* had a peak in morning and evening (Fig. 4), around sunrise and sunset. We can speculate that these are the periods when sunlight was still available, but at limited amounts. This effect may cause an optimal situation for having both $NO_3$ and NO participating in the oxidation process. This is supported by the high N-atom content of the main species in this factor. Meanwhile, some other factors showed an opposite trend to UB2-1, namely minima during morning and evening, often with a strong nighttime peak and a smaller daytime increase. Some of the most prominent factors with such behavior were *SC2-3: MT monomers 2*, *UB2-2: MT monomers 2*, *UB2-3: sesquiterpene 1*, and *UB3-1: sesquiterpene 2*. $NO_3$ was identified as the main oxidant for these factors based on the mass spectra and the high nighttime signals, but the local maxima around noon is surprising. Saarikoski et al. (2023) did estimate that $NO_3$ would have a small daytime maximum, likely due to the sinks not being fast enough to fully overwhelm the very high formation rates from high $O_3$ and $NO_2$ during this time. We cannot determine to which extent the diurnal variation of $NO_3$ influences these diurnal patterns. As was the case also in many situations discussed above, we are often unable to separate if an increase is due to an enhanced source strength or a decrease in competing reaction pathways.

**Comparison to previous research**

HOM data from Helsinki show similarities with previous studies done on ambient HOM data in urban as well as rural environments. Yan et al. (2016) investigated HOM formation pathways at a boreal forest site (SMEAR II station, Hyytiälä, Finland) located approximately 190 km from Helsinki and 50 km from the closest city – Tampere (with population approximately 250 000). A factor "*Nighttype type-2"*, obtained from PMF analysis by Yan et al. (2016) contained MT-derived dimers formed by $NO_3$ and $O_3$ oxidation and $RO_2$ termination. That factor mostly consisted of $C_{20}H_{31}O_{10-18}N$ (40%) and $C_{20}H_{32}O_{10-17}N_2$ (20%) suggesting that the dimers detected in Hyytiälä and in Helsinki (both stations) have the same formation pathways, even though these measurement sites represent different rural and urban environment. Despite relatively similar precursors and formation pathways, far fewer similarities are found between the mass spectra of MT-derived monomer factors at these three sites. This suggests, as also mentioned above, that monomer formation pathways are much more diverse compared to dimer formation. Still, a comparison of our results with other studies done in rural environments (Massoli et al., 2018; Kürten et al., 2016) showed clearly lower resemblance between MT-derived dimers, which is likely a result of different biome types in their studies (isoprene-dominated south east US and rural agricultural site in Germany respectively) compared the ones conducted in Finland.

In recent years, more research on condensable vapor formation has been conducted in urban environments heavily influenced by $NO_x$ (Yan et al., 2022; Guo et al., 2022b; Liu et al., 2021; Nie et al., 2022; Zhang et al., 2022). Unlike forest environments, where the fraction of nitrogen-containing HOM is similar to the fraction of HOM without nitrogen atoms (Yan et al., 2016; Massoli et al., 2018), condensable vapor composition in Chinese megacities is dominated by nitrogen-

containing compounds, which represent approximately 60-85% of all measured condensable vapors (Guo et al., 2022b; Liu et al., 2021; Nie et al., 2022; Zhang et al., 2022). This strong influenced by $NO_x$ was also observed in the present study at both stations in Helsinki. In addition, the majority of key compounds in *SC1-5: nitrophenol & aliphatic* are also listed as main compounds in factors originating from aliphatic AVOCs detected in Nanjing (*Aliph-OOM*) (Liu et al., 2021) and in Beijing (*aliphatic OOMs*) (Guo et al., 2022b). However, depending on the time of the year, the main precursors for condensable vapors in cities in China are either AVOCs or a mix of AVOCs and BVOCs (Guo et al., 2022b; Liu et al., 2021; Nie et al., 2022). This is clearly different compared to Helsinki, where BVOC-derived vapors were more abundant. This dissimilarity is likely due to the AVOC:BVOC ratio being much larger in Chinese cities due to closer proximity of much larger areas with anthropogenic emissions. In contrast, Helsinki AVOC:BVOC is much smaller due to larger BVOC emissions from abundant vegetation in the close surroundings. It is also important to notice that most studies of condensable vapors in Chinese cities (Guo et al., 2022b; Liu et al., 2021; Nie et al., 2022) analyzed much smaller mass range (200-400 Th or 250-400 Th), which corresponds to range 1-2 here. In our study, range 1 is the only mass range in which we find the dominant influence of anthropogenic precursors. Brean et al. (2019) also showed that MT-derived dimer concentrations were approximately 50 times lower than MT-derived monomers in Beijing, likely due to both small MT emissions and suppression of dimer formation by NO.

### 3.3. Implications for air quality

Ambient air pollution was recognized as the largest environmental health risk and one of the top risk factors for the loss of healthy years (Lim et al., 2012; Anderson et al., 2012; Cohen et al., 2017). Premature deaths caused by ambient air pollution are linked to particular matter (PM) (Cohen et al., 2017; WHO, 2021), both due to short-term (Pope and Dockery, 2012) and long-term exposure (Burnett et al., 2014). In many environments, including different urban areas, PM is dominated by secondary aerosol formed from condensable vapors, including HOM. HOM and other condensable organic vapors impact not only PM concentration but also the chemical composition of SOA and, consequently, aerosol properties like toxicity. For example, in a recent study with human alveolar epithelial cells and human monocyte cells, the organic compounds and the aging of the aerosol were major drivers of the cell level toxicity of aerosol (Hakkarainen et al., 2022).

In this work, we found that the majority of low-volatility condensable vapors in Helsinki were impacted by both biogenic and anthropogenic precursors, despite high local anthropogenic emissions. The VOC precursors themselves were mostly of biogenic origin, i.e. BVOC, but the oxidation process was strongly perturbed by anthropogenic activity, particularly by $NO_x$. While detailed similarities in mass spectra of factors were often small between the close-by sites studied here, most observed compounds at both stations were ONCs. Previous studies have shown that $NO_x$ can change the yield of SOA formation during VOC oxidation (Mutzel et al., 2021; Jaoui et al., 2013; Ng et al., 2017), though this effect may be not as clear to observe in ambient measurements (Yan et al., 2022). In the smaller m/Q ranges studied in this work, the influence from AVOC was larger, but we cannot deduce the impact of these factors on SOA formation due to their semi-volatile nature. Nevertheless, our results indicate that in Helsinki, and likely in other biologically influenced urban areas, anthropogenic emissions affect HOM formation and composition most strongly by the participation of $NO_x$ in the (B)VOC oxidation. That influence will be propagated to the SOA, both concerning the composition as well as the effective yield of SOA from the BVOC oxidation, but quantifying the ultimate impact on either of these will require further studies.

## 4. Conclusions

We measured the composition of condensable vapors, HOM, during late spring at two stations separated by 900 m in different sub-environments in Helsinki, a city with considerable biogenic influence from trees. We compared HOM composition and formation pathways at the two sites, an urban background station and a street canyon, using PMF analysis to separate the complex data into covarying compound groups. We found that the majority of the HOM originated from BVOCs at both locations, despite them being dominated by AVOC emissions (Rantala et al., 2016; Saarikoski et al., 2023). However, we did observe a strong anthropogenic influence on the HOM formation, due to the elevated $NO_x$ concentrations at both stations, which is consistent with previous studies conducted in urban environments (Guo et al., 2022b; Liu et al., 2021; Nie et al., 2022). The PMF factors, and their temporal behavior, were surprisingly different between the two sites, considering their relatively close proximity. Monoterpene-derived dimers were the compound groups that correlated best between the sites. On the contrary, at the street canyon site we observed a factor corresponding partly to AVOC-derived factors found in Chinese megacities (Guo et al., 2022b; Liu et al., 2021; Nie et al., 2022). The lack of a similar factor in the PMF solution from the urban background station highlights that HOM composition at two nearby sites in an urban environment can differ noticeably depending on the local anthropogenic influences. To a large extent, we expect this difference to be driven by differences in the environmental conditions, leading to distinct oxidation products even when the same VOC molecule becomes oxidized, due to competition between both oxidants and $RO_2$ terminators.

Our work indicates that when analyzing and discussing the impact of HOM on SOA and air quality in urban environments, we need to keep in mind the spatial inhomogeneity of urban areas in the HOM composition and formation mechanisms. Thus, a more detailed investigation of the formation and composition of HOM in a variety of different urban sub-environments would be beneficial. Additionally, our findings are restricted to a short and biologically active period, hence follow-up research on seasonal changes is needed. Finally, we recommend that future mass spectrometric studies in urban area employ devices with resolving power above 5000 Th/Th, as the mass spectra are extremely complex and thus even peak identification can be a major challenge.

**Data availability**

All data presented in this manuscript will be available in open repository before the final version of manuscript is completed.

**Author contribution**

The main ideas were formulated by OG, HT, JKo, JVM, MS, MDM TR, TP, MK and the results were interpreted by MOk, OG, PP, and ME. TR, HK, OG, HT prepared measurement methodology and OG, MOl, JKa, HH, and HK contributed to data collection. MOk performed the data analysis and YZ supported it. OG and EH supervised the project. HT, MDM, and TP made a funding acquisition. MOk visualized data and prepared the manuscript with contributions from OG, ME. All the authors reviewed and commented the manuscript.

Competing interests

The authors declare that they have no conflict of interest.

Acknowledgments.

This research was supported by the Regional innovations and experimentations funds AIKO (project HAQT, AIKO014), Business Finland (CITYZER project, Tekes nro: 3021/31/2015 and 2883/31/2015), Pegasor Oy and HSY, Academy of Finland (grant nos 273010, 307331, 310626, 311932, 318940, 1325656, 326437, and ACCC flagship grant no. 337549, 337552, 337551), Healthy Outdoor Premises for Everyone (HOPE), Urban Innovation Actions, Regional development funds, The Technology Industries of Finland Centennial Foundation via project Urbaani ilmanlaatu 2.0, European Commission via Horizon Europe project "Non-CO2 Forcers and their Climate, Weather, Air Quality and Health Impacts, FOCI" (101056783), Faculty of Science 3-year grant (75284132), Tampere University of Technology graduate school, European Research Council (ERC) project ADAPT (grant no. 101002728), European Union Horizon 2020 research and innovation programme (grant no 101036245, RI-URBANS; grant no. 821205, FORCeS, and ERA-PLANET project SMURBS, 689443).

We would like to thank the people who took care of instruments and helped with measurements at the SMEAR III (Pekka Rantala, Erkki Siivola, Pasi Aalto, Petri Keronen, Frans Korhonen, Tiia Laurila, Lauriane Quélérver, Tuuli Lehmusjärvi, Deniz Kemppainen) and the HSY Mäkelänkatu site (Anssi Julkunen, Anders Svens, Harri Portin, Taneli Mäkelä, Tommi Wallenius, Anu Kousa).

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
