# Peer review of "Influence of anthropogenic emissions on the composition of highly"

_EGUsphere, 2023_

## Author Response (AR1)

Response to reviewer comments for manuscript 'Influence of anthropogenic emissions on the composition of highly oxygenated organic molecules in Helsinki: a street canyon and urban background station comparison'

We thank the reviewers for their valuable comments and suggestions that have helped to improve our manuscript. Here, we present answers to each of their comments in blue. Reviewers' comments are quoted as black text, while the text from the manuscript is marked with *grey italics* and the changes in the text are marked in *red italics*.

**Reviewer 1**

Okuljar et al. present measurements of HOMs and other condensable vapors from measurements at two sites in 1 km distance from another in Helsinki. The manuscript is well written and uses state-of-the-art scientific methods.

Given that there is still a lot that the atmospheric chemistry community does not know about the formation of HOMs/SOA, these observations in a suburban area provide an important addition to the literature. What I find the most interesting and striking point of this study is the strong difference between the composition and diurnal behavior of observed compounds during the same time, although the two sites are spatially so close to each other. This points to an important lesson for the atmospheric chemistry community regarding the (non-)representativeness of a single observation site for a whole metropolitan area, and generally to a very localized inhomogeneity of emissions, reaction processes and products. Therefore, I wish this point would be brought across more strongly, perhaps graphically. Thus see my comments below.

We thank the reviewer for all their useful input. Concerning the emphasis on inhomogeneities, we now added the following sentence to the end of the abstract to make this point clearer:

**This further suggests that studies should be careful when extrapolating single-point measurements in an urban setting to be representative for district or city scales.**

I recommend this article for publication in ACP after the following comments have been addressed:

**General comments:**

1. Did you see any evidence for cooking emissions contributing to any of the observed PMF factors? E.g. Zotter et al. (2014, https://doi.org/10.1002/2013JD021114, 2014) mention that at least 25% of the non-fossil SOA is from cooking.

We do not observe cooking emissions contributing to PMF factors. Although, cooking emission can have important contribution to non-fossil SOA (Zotter et al., 2014), previous studies in Helsinki have not identified any factor of SOA associated with cooking (Timonen et al., 2013). It is likely due to smaller population density and sparser restaurant distribution in measurement area of Helsinki than in Los Angeles basin. Since the cooking contribution was not observed for particles, it is also not surprising that we do not expect to find its contribution to such signatures in the condensable vapors measured in this study.

The contribution of emissions from a near-by coffee roastery to SOA has previously been observed at the urban background station (Timonen et al., 2013), and we tried to investigate whether we can see any contribution to condensable vapors. We did not find any factor influenced by the coffee

roastery but were also not able to detect any contribution to SOA during our campaign (based on AMS measurement from corresponding time). The absence of coffee roastery factor in our data is likely due to the very low contribution (1%) and high variance of this factor (Timonen et al., 2013).

2. You report that only few PMF factors correlated with each other between the two sites. I wonder if some of the reason may be that the air transported from one place to the other by prevailing winds takes some time, so that the correlation would be shifted in time, and would require a method like time-warping to find correlations? Also, was the PMF conducted separately for the time where there is overlapping data between the two sites? I wonder if there could be a bias if the latter is not the case, since both sites were measured for a longer time separately than simultaneously.

We agree that there could be a time shift between factors which we are not accounting for, thus we had calculated cross-correlation for our factors. The Pearson's coefficient increased only by a few hundredths while using cross-correlations, which is a negligible difference, and thus we have not discussed it in the manuscript. Time warping methods could be suitable to account for differences in the speed of processes influencing condensable vapor formation and transport, but we find time-warping methods risky to use for calculating correlations as it may greatly overestimate them. We tried dynamic-time warping with the time constrained, and it improves correlations of factors that should not correlate with each other as they originate from different sources on different stations (for example Fig. R1). We think that dynamic time warping should be only used when we are sure that measured condensable vapors at both sites were formed from VOCs emitted from the same source. Unfortunately, we cannot make this assumption for our data.

Another aspect to keep in mind is that condensable vapors are very short-lived species; for average condensation sink equal to  $0.005 \text{ s}^{-1}$  (Okuljar et al., 2021) the estimated lifetime of HOM is no longer than 2 min (Bianchi et al., 2019). With a 900 m distance between the stations and median wind speed of approximately 3 m/s, during most measurement time we do not expect to observe exactly the same molecules at both stations. The similarities of condensable vapors composition come rather from the same processes of HOM formation. Additionally, based on previous study on the dispersion mechanisms for particles and trace gases from a highway in Helsinki (Pirjola et al., 2006), we can estimate that compounds that could survive the transportation, would most likely have too small concentration to detect them on the downwind station. Lastly, we would like to point out that the time resolution of the data used for PMF is 30 minutes, thus any species transported with wind should largely end up in the same data point. Since we do not expect to be able to measure the same molecules at both stations, we did not run the PMF for the time when we had simultaneous measurement but rather focused on describing each station separately.

While looking at correlations between PMF factors, we noticed that one and half day of data was removed from the correlation analysis by accident. After adding the missing data, the Pearson coefficients for correlations between PMF factors did not change or changed negligibly. Nevertheless, the Pearson coefficient for correlation between factor SC2-1 and UB1-6 decreased from 0.7 to 0.66, thus we removed the time-series of these factors from Fig 5. We also applied changes to Fig. S12.

Figure R1. Time series of not correlating factors (R=0.37) SC2-2 and UB2-3 before (top) and after (bottom) applying dynamic time warping. The x-axis is a time axis of ordered list of datapoints starting from 14 May 2018 at 16:00. After applying dynamic time warping, the actual time stamps are disregarded but the order of datapoints is kept.

Inspired by this question, we decided to add an estimation of the ceiling correlation, the highest correlation expected between stations calculated from sulfuric acid time-series, for times series of condensable vapors between stations:

Tables 1 and S3 present the most plausible interpretation of selected and not selected factors, respectively. For each factor, we propose VOC precursors, oxidants, and terminators, which were most likely to influence the formation of species in this factor. We also specify an hour of the day when factor's signal reached its maximum as well as the contribution of this factor to the total signal both within its own m/Q sub-range and within the full analyzed range (200-650 Th). See Table 1 caption for a more detailed description of how to read the table. The findings and implications are discussed below. While discussing time series correlations between factors from both stations, it is important to keep in mind that they cannot be ideal. We estimated that the highest correlation between stations for condensable vapors is approx. 0.88 which is a correlation between concentrations of a compound that is mostly produced in the same pathway on both sites: sulfuric acid (SA) (Fig. S15).

*Figure S15. Time series of sulfuric acid measured at street canyon (SC)(red) and urban background (UB)(black) for the overlapping time of measurement.*

3. How deep is the "street canyon", i.e., how high are the surrounding buildings? If they are high: Do you see any evidence that the street canyon site has more stagnant air, i.e., that emissions spend more time there so that local emissions reach higher oxidation states, vs. a more well-ventilated suburban site, where observed HOMs may stem from longer range transport? I think the discussion lacks a little bit of consideration that local emissions and reactants may not be the only reason for the differences observed between the two sites. Please discuss meteorology/transport.

We added a short description of the street canyon and a paragraph discussing pollution transport at the stations:

Station for Measuring Ecosystem-Atmosphere Relations (SMEAR III, 60°12′10.4′′ N, 24°57′40.2′′ E) (Fig. 1). The HSY supersite is located at a street canyon, less than a meter from Mäkelänkatu street (around 28 000 vehicles/weekday) (Kuuluvainen et al., 2018). The street canyon is 42 m wide and the height of buildings on the both sides of the supersite is 19 and 16 m, leading to the average height-to-width ratio of 0.45 (Järvi et al., 2023). It contains a pavement and three lines of road for both directions separated by the two tram lines and trees.

The detailed meteorological description of the transport of pollutants at both stations with the emphasis on the mechanism affecting this transport at the street canyon is presented by Järvi et al. (2023). In summer the atmosphere is very stable during nighttime and very unstable during daytime at the urban background station. The meteorological conditions at the urban background station (Fig. 2) resemble the one described by Järvi et al. (2023) for summer, which suggests limited vertical mixing of the atmosphere during nighttime and very well-mixed lower atmosphere during daytime during our measurement. At both stations, during the warm period the mechanical and thermal mixing is stronger than during cold periods resulting in conditions more favorable for pollution dispersion (Järvi et al., 2023). At the street canyon, not only mean wind but also the turbulent mixing is important for the transport of pollutants. This may suggest that even though the air is not stagnant, the pollutants are not efficiently transported from the street canyon.

4. I suggest adding a graphical depiction of the amount of overlap between the observed signals between the two sites since this is such an important message. Maybe a pie chart showing the fraction of total ions (sum of both sites) only observed at SC, fraction only observed at UB, fraction observed at both sites? E.g. similar to Fig. 2 of https://doi.org/10.1021/acs.est.2c07260.

We agree that it would be beneficial for the paper to make a visual comparison between stations. Nevertheless, we cannot think of a way to present it without the risk of being misleading. We were not able to identify all peaks in our mass spectra, sometimes due to low signals, sometimes due to overlapping ions. As such, the choice of whether a given ion was in fact detected or not becomes ambiguous for a large number of peaks. The current comparison in the paper reflects all relations between the two stations that we were confident to interpret from our data. The inability of comparing the results from both stations in a broader sense is one of the reasons why we recommend a follow-up study that uses a mass spectrometer with a higher mass resolution than the one used in this study.

**Specific comments:**

1.44: apart from the listed sources, I suggest to mention cooking since it is also a strong source of anthropogenic VOCs/ condensable vapors.

**We added cooking as a source of VOC:**

The sources of anthropogenic emissions consist of traffic, **cooking**, industrial processes and production of customer goods, and volatile chemical products (VCP) (Li et al., 2022; Koppmann, 2007; Watson et al., 2001).

Fig. 1: It would be helpful to add a wind rose to see which of the two stations is up-/downwind of each other.

This is a good suggestion and we think that a wind rose could also help understanding the atmospheric conditions at our stations in general, thus we added it to the Fig. 2:

Figure 2 presents diurnal variations of measured variables that can influence HOM formation pathways: global radiation, ambient temperature (T), and concentrations of  $O_3$ , NO, and NO2 as well as the wind direction measured during this campaign.